# Searching for Chemical Agents Suppressing Substrate Microbiota in White-Rot Fungi Large-Scale Cultivation

**DOI:** 10.3390/microorganisms12061242

**Published:** 2024-06-20

**Authors:** Audrius Maruška, Rūta Mickienė, Vilma Kaškonienė, Saulius Grigiškis, Mantas Stankevičius, Tomas Drevinskas, Olga Kornyšova, Enrica Donati, Nicola Tiso, Jurgita Mikašauskaitė-Tiso, Massimo Zacchini, Donatas Levišauskas, Ona Ragažinskienė, Kristina Bimbiraitė-Survilienė, Arvydas Kanopka, Gediminas Dūda

**Affiliations:** 1Instrumental Analysis Open Access Centre, Vytautas Magnus University, Vileikos St. 8, LT-40444 Kaunas, Lithuania; ruta.mickiene@vdu.lt (R.M.); vilma.kaskoniene@vdu.lt (V.K.); mantas.stankevicius@vdu.lt (M.S.); tomas.drevinskas@gmail.com (T.D.); olga.kornysova@vdu.lt (O.K.); nicola.tiso@vdu.lt (N.T.); jurgita.mikasauskaite-tiso@vdu.lt (J.M.-T.); donatas.levisauskas@ktu.lt (D.L.); kristina.bimbiraite-surviliene@vdu.lt (K.B.-S.); arvydas.kanopka@vdu.lt (A.K.); gediminas.duda@vdu.lt (G.D.); 2JSC Biocentras, Pagirių St. 1P, LT-14118 Pagiriai, Lithuania; biocentras@biocentras.lt; 3National Research Council, Area Della Ricerca di Roma 1, Via Salaria Km 29,300, Monterotondo, 00015 Rome, Italy; enrica.donati@cnr.it (E.D.); massimo.zacchini@cnr.it (M.Z.); 4Process Control Department, Kaunas University of Technology, Studentų St. 50, LT-51368 Kaunas, Lithuania; 5Botanical Garden of Vytautas Magnus University, Ž. E. Žilibero 6, LT-46324 Kaunas, Lithuania; ona.ragazinskiene@vdu.lt

**Keywords:** *Irpex lacteus*, *Pleurotus ostreatus*, *Pleurotus eryngii*, economical substrate disinfection, fungi cultivation, food source, bioremediation

## Abstract

Edible fungi are a valuable resource in the search for sustainable solutions to environmental pollution. Their ability to degrade organic pollutants, extract heavy metals, and restore ecological balance has a huge potential for bioremediation. They are also sustainable food resources. Edible fungi (basidiomycetes or fungi from other divisions) represent an underutilized resource in the field of bioremediation. By maximizing their unique capabilities, it is possible to develop innovative approaches for addressing environmental contamination. The aim of the present study was to find selective chemical agents suppressing the growth of microfungi and bacteria, but not suppressing white-rot fungi, in order to perform large-scale cultivation of white-rot fungi in natural unsterile substrates and use it for different purposes. One application could be the preparation of a matrix composed of wooden sleeper (contaminated with PAHs) and soil for further hazardous waste bioremediation using white-rot fungi. In vitro microbiological methods were applied, such as, firstly, compatibility tests between bacteria and white-rot fungi or microfungi, allowing us to evaluate the interaction between different organisms, and secondly, the addition of chemicals on the surface of a Petri dish with a test strain of microorganisms of white-rot fungi, allowing us to determine the impact of chemicals on the growth of organisms. This study shows that white-rot fungi are not compatible to grow with several rhizobacteria or bacteria isolated from soil and bioremediated waste. Therefore, the impact of several inorganic materials, such as lime (hydrated form), charcoal, dolomite powder, ash, gypsum, phosphogypsum, hydrogen peroxide, potassium permanganate, and sodium hydroxide, was evaluated on the growth of microfungi (sixteen strains), white-rot fungi (three strains), and bacteria (nine strains) *in vitro*. Charcoal, dolomite powder, gypsum, and phosphogypsum did not suppress the growth either of microfungi or of bacteria in the tested substrate, and even acted as promoters of their growth. The effects of the other agents tested were strain dependent. Potassium permanganate could be used for bacteria and *Candida* spp. growth suppression, but not for other microfungi. Lime showed promising results by suppressing the growth of microfungi and bacteria, but it also suppressed the growth of white-rot fungi. Hydrogen peroxide showed strong suppression of microfungi, and even had a bactericidal effect on some bacteria, but did not have an impact on white-rot fungi. The study highlights the practical utility of using hydrogen peroxide up to 3% as an effective biota-suppressing chemical agent prior to inoculating white-rot fungi in the large-scale bioremediation of polluted substrates, or in the large-scale cultivation for mushroom production as a foodstuff.

## 1. Introduction

Basidiomycetes are considered to be the most complex and evolutionarily advanced members of the fungal kingdom, playing vital roles in carbon cycling and as symbiotic partners with other organisms [1]. They are a highly valuable food source, and are increasingly important in medicinal applications due to their diverse bioactivities and potential to be promising and effective agents for various purposes (antihyperlipidemic, antibacterial, antifungal, antiviral, cytotoxic, immunomodulating, and antioxidant) [2,3]. They are known to produce natural antibiotics with antimicrobial, anticancer, and antioxidant activities [4,5]. Basidiomycetes possess exceptional abilities to degrade lignocellulose, making them potentially useful in exploring lignocellulosic biomass for the production of fuel ethanol and other chemicals, as well as in bioremediation processes [6,7]. Overall, it is challenging to determine whether Basidiomycetes are more important as a food source, medicinal source, or as a decomposer, as each role contributes significantly to human well-being and ecosystem functioning in different ways. However, their importance in providing nutritious food, potential therapeutic compounds, and essential ecosystem services highlights the multifaceted significance of Basidiomycetes in both human and environmental contexts.

Certain species of edible mushrooms have unique metabolic capabilities that allow them to break down various organic pollutants and even absorb and accumulate some heavy metals. The ability to accumulate certain metals can be utilized for cultivating mushrooms enriched with specific metals, such as selenium [8], which could help alleviate selenium deficiency in the diet. These fungi can break down complex organic compounds through enzymatic activity, converting pollutants into simpler, less harmful substances. The unique enzymatic activity of fungi contributes to their effectiveness in bioremediation by degrading toxic organic compounds, converting pollutants into simpler, less harmful substances [9].

Applying bioremediation to fungi or microorganisms is a useful and environmentally friendly technique for the reduction in pollutant levels in almost any media. Published research shows that the use of bioremediation allows us to reduce the level of pollution in wastewater [10,11], polluted soil [12,13,14], tannery effluent [15], railway sleepers polluted by creosote [16,17], and pharmaceuticals [9,18], etc. Fungi have been widely used in bioremediation due to their ability to adapt to different environmental conditions [18,19,20]. Some of the key factors that contribute to the adaptation of fungi in bioremediation include the following. (i) Enzymatic activity: Fungi can produce a variety of enzymes that can degrade a wide range of organic pollutants. This enzymatic activity is an important factor that contributes to the adaptation of fungi to bioremediation conditions [21,22,23]. (ii) Versatility: Fungi are able to grow in a wide range of environmental conditions, including extreme temperatures, pH levels, and high salt concentrations. This versatility allows them to adapt to different bioremediation conditions [24,25,26]. (iii) Cell wall composition: The composition of the fungal cell wall is another important factor that contributes to their adaptation to bioremediation conditions. Fungi have the ability to modify their cell wall structure in response to changes in the environment, which can help them adapt to different conditions [27,28]. (iv) Nutrient utilization: Fungi have the ability to utilize a wide range of nutrients, including those present in contaminated soils and liquid media. This ability allows them to adapt to the specific nutrient conditions found in contaminated environments [29]. (v) Mutualistic relationships: Some fungi have mutualistic relationships with other microorganisms, meaning that both species benefit from their interaction, such as bacteria, which can contribute to their adaptation to bioremediation conditions. For example, fungi can provide nutrients and other growth factors to bacteria, which can, in turn, help to degrade pollutants [30].

In spite of the promising results of bioremediation, this process is not so popular at the industry level, as each case of bioremediation requires a lot of specific customization, which investors do not like as they desire less specific, widely applied technologies [31]. The achievement of certain initial sterilities to suppress unwanted bacteria and microfungi and initiate the growth of the desired fungi or microorganisms is a challenge when working with high amounts of polluted material or when cultivating mushrooms on a large-scale for production as a foodstuff. The suppression of the growth of bacteria and microfungi is important as an initial stage of bioremediation, while the fungi inoculum will adapt to the new environment and will start to bioremediate the polluted material. In spite of the complex and diverse interactions of bacteria and fungi in various environments, including soils, water, and the human body [32,33,34,35], some of these interactions may not be mutualistic, but rather may be competitive or even parasitic, where one species benefits at the expense of the other [30]. Bacteria and fungi may compete for the same limited resources, such as nutrients and space, especially when not-sterile bioremediated material is used and the process is intended to be carried out under natural conditions. These interactions can lead to the inhibition or suppression of the growth of one or both species, which can be a crucial issue for the successful bioremediation process. 

In order to reduce environmental pollution, many scientists seek to find ecological ways to reduce chemical or microbiological pollution. As previously mentioned, fungi can be used to reduce chemical pollution in various substrates through mycoremediation [19,36], certain plants [37,38], or even by combining phytoremediation with mycoremediation [17,20]. When it comes to reducing microbiological pollution, plant biocides and phytoncides [39], or even bacteriocins [40], released by bacteria can be used, although the latter are more commonly applied in the food or pharmaceutical industry, but are not commonly applied in environmental protection. Good antimicrobial activity of ZnO, TiO_2_, CuO, silver, and gold nanoparticles produced in an eco-friendly manner were recently discovered [41,42,43,44], and may have future prospectives for the reduction in microbial pollution in different areas.

Depending on the genus and species of bacteria, they exhibit different sensitivities to various drugs and chemical agents. Probably the major factor determining bacterial sensitivity or resistance is the structure of the cell wall. Other factors, such as genetic makeup, efflux mechanisms, enzymatic degradation, biofilm formation, and target site mutations, also play significant roles [45]. According to the structure of their cell walls, bacteria are classified as Gram-negative or Gram-positive. However, some bacteria, such as mycobacteria, are not reliably stained by the Gram method due to the high lipid content in their cell walls, and some scientists suggest classifying these bacteria as high G+C Gram-positives [46]. Mycobacteria have a unique and complex cell wall rich in mycolic acids, which make them waxy and impermeable to many antibiotics [47] and other drugs [48], including common disinfectants like H_2_O_2_ [49]. However, some laboratory-scale studies show the sensitivity of certain mycobacterial strains to natural substances, such as ginger essential oil [50], or eco-friendly disinfectants, including a phenolic-based disinfectant or a quaternary ammonium-based disinfectant [48].

According to European Commission Regulation No. 1272/2008 [51], creosote (CAS No 90640-85-0) is classified as a 1B category cancerogenic material, and may cause cancer by inhalation (hazard code H350). The utilization of used creosote-soaked sleepers is a huge problem not only in Lithuania, but also throughout Europe and the world, as the only official method is incineration. A huge drawback of this method is the high amount of toxic substances (phenols, phenanthrene, acetone, and butanol). These substances have a negative impact not only on the environment, but also on human health, contributing to the emergence and development of various diseases, including cancer. Therefore, due to environmental requirements, the burning of sleepers is limited, and large amounts of used sleepers are stored in waste sites. Therefore, research related to the utilization of used wooden sleepers is highly relevant. 

The aim of the present study was to find selective chemical agents that suppressed the growth of microfungi and bacteria which were isolated from the substrate for large-scale cultivation as a foodstuff or as hazardous waste intended for bioremediation, but also to find agents that simultaneously did not suppress white-rot fungi during cultivation. It is important to find simple, cheap, and effective means for the disinfection of bioremediation substrates when the usual heat sterilization method is not economically beneficial due to the high amounts of the substrate required to be treated.

The proposed substrate pre-treatment method could be attractive for growing edible mushrooms, where proper pasteurization of the substrate is essential for successful mushroom cultivation. It requires no special equipment, such as pasteurization tunnels or steam chambers, which are typically used for eliminating competing microorganisms and pathogens. Additionally, the process is faster than hot water soaking or composting.

## 2. Materials and Methods

### 2.1. Chemical Materials and Growth Media

The means used for disinfection in this study were the following: charcoal (JSC “Monada LT”, Lithuania), phoshpgypsum (CaSO_4_·1/2 H_2_O, SC Lifosa, Kėdainiai, Lithuania), dolomite powder Dirvitas (composition: 30% CaO, 20% MgO, 51% CaCO_3_, 42% MgCO_3_; SC Dolomitas, Akmenė, Lithuania), gypsum (Meyercordt GmbH, Bad Salzuflen, Germany), calcium lime 80 in the form of hydrated lime (Krasnoselskstroimaterialy, Belarus), ashes of wood pellets (JSC “Strielčių granulės” Strielčiai, Lithuania), hydrogen peroxide (50%, Eurochemicals, Kuprioniškės, Lithuania), potassium permanganate (chem. pure, JSC Valentis, Vilnius, Lithuania), and sodium hydroxide (chem. pure, S.r.o. Reachem Slovakia, Bratislava, Slovakia). 

The following media were used for the growing of organisms: LB Medium (Lennox), Malzextrakt-bouillon, and LB agar (all from VWR International GmbH, B.D.H., Vienna, Austria).

### 2.2. Tested Organisms

*Bacteria Bacillus licheniformis*, *Bacillus mycoides*, *Bacillus subtilis*, *Bacillus thuringiensis*, *Candida* spp., *Klebsiella variicola*, and *Pseudomonas fluorescens* were isolated from the biohumus (JSC Biohumus & Soil, Rokiškis, Lithuania) and used wooden railway sleeper chips polluted with creosote [52,53]. Bacteria were identified using a DNA sequence and the databases of the National Center for Biotechnology Information [54]. Bacteria were grown in —LB Lennox (Carl Roth GmbH, Karlsruhe, Germany) media in incubator ICF120 (AgroLab, Altavilla Vicentina, Italy) at 37 °C temperature for 48 h. Three rhizobacteria from the rhizosphere of root vegetables, namely *Azotobacter vinelandii* Lipman (ATCC 478), *Bacillus megaterium* de Bary (ATCC 14581), and *Bacillus mojavensis* Roberts et al. (ATCC 51516), were used in the tests. 

Microfungi *Acremoniella verrucose*, *Aspergillus fumigatus*, *Aspergillus niger*, *Candida* spp., *Chrysosporium merdarium*, *Cryptococcus laurentii*, *Cryptococcus neoformans*, *Fusarium moniliforme*, *Memnoniella echinate*, *Myrothecium verrucaria*, *Penicillium funiculosum*, *Penicillium paxilli*, *Rhizomucor pusillus*, *Trichoderma harzianum*, *Trichophyton rubrum*, and *Ulocladium chartarum* were also isolated from the biohumus and hazardous waste, i.e., wooden railway sleeper chips. Tested microfungi were grown in the malt extract at 28 °C for 7–10 days. The identification of the microfungi morphology was performed on the 7-day-old cultures using light microscopy, on a microscope Novex Holland K-range (Arnhem, The Netherlands). The identification of the isolated microfungi was conducted as per the guidelines and general principles of fungal classification [55,56,57].

White-rot fungi *Irpex lacteus*, *Pleurotus ostreatus*, and *Pleurotus eryngii* isolated from Lithuania microbiota in our previous study [22] were grown in the malt extract at 28 °C for 7–10 days.

The mentioned strains were isolated from the biohumus and hazardous waste, as the prospect of this study is an application of white-rot fungi for large-scale cultivation in natural nonsterile conditions for different purposes. The experimental plan and idea of this study is figured in Figure 1.

### 2.3. Evaluation of the Impact of Bacteria and Yeast on the Growth of Microfungi and White-Rot Fungi

For the evaluation of the impact of bacteria isolated from the biohumus and waste intended for bioremediation (*B. licheniformis*, *B. mycoides*, *B. subtilis*, *B. thuringiensis*, *K. variicola*, and *P. fluorescens*) and rhizobacteria (*A. vinelandii*, *B. megaterium*, and *B. mojavensis*), and yeast (*Candida* spp.) on the growth of white-rot fungi and microfungi, 100 µL of suspension was spread onto the malt extract agar in 90 mm Petri dishes with a sterile Drigalski spatula. Then, a piece (appr. 10 × 10 mm in size) of white-rot fungi mycelium was cut with sterile instruments and placed in the middle of a Petri dish. An inoculum of microfungi was carried onto the center of the Petri dish contaminated with bacteria using a sterile loop. Parallelly, control Petri dishes (without a spread of bacteria) with specific fungi or microfungi were prepared for comparison. The clear zones were measured around the fungi mycelium (in mm), and its growth or decline was evaluated on the 2nd and 9th days of incubation at a temperature of 26–28 °C. All experiments were repeated three times. 

### 2.4. Evaluation of Suppression of Organism’s Growth by Different Chemicals

The impact of several chemicals on the growth of bacteria, microfungi, and white-rot fungi were tested. Bacteria and microfungi were cultivated in the liquid media (in LB Medium and malt extract, respectively), while the cell density reached 0.5 McF units measured using a densitometer DEN-1 (Biosan, Riga, Latvia). Then, the cultivation 100 µL of suspension was spread onto the solid LB or malt extract agar in 35 mm Petri dishes with a sterile Drigalski spatula. Then, 15 µL of NaOH solution (0.1–3.0%), H_2_O_2_ (1.5 and 3.0%), or KMnO_4_ (0.01–1.0%) were dropped onto the agar with the tested bacteria or micromycete. A certain amount of lime (20 ± 0.1 mg), dolomite powder (80 ± 0.1 mg), charcoal (50 ± 0.1 mg), wood ashes (15 ± 0.1 mg), gypsum (20 ± 0.1 mg), or phosphogypsum (60 ± 0.1 mg) powder was added on the surface of the media with a specific microorganism when solid materials were tested. Then, plates were incubated at 26–28 °C for 7–10 days for microfungi and at 37 °C for bacteria for 1–2 days. The growing microfungi colonies were assessed visually on 7th to 10th day of development, while the bacteria were assessed visually on the 2nd or 3rd days.

White-rot fungi were tested for sensitivity to different concentrations of NaOH (0.1–3.0%) and KMnO_4_ (0.01–1.0%). The inoculum of prepared white-rot fungi was homogenized, and then 100 µL of suspension was spread onto solid malt extract, and 15 µL of NaOH or KMnO_4_ solutions was dropped onto it. The impact of lime and ashes were assessed in the same way described above. The plates were incubated in an incubator (Biosan, Riga, Latvia) at 26–28 °C. Growing colonies of white-rot fungi were assessed visually on the 7th to 10th day.

In parallel, control Petri dishes (without the addition of chemicals) with a specific organism were prepared for comparison. After a proper incubation time, the growth of organisms was evaluated microscopically using a comparison with the control. All experiments were repeated three times.

## 3. Results and Discussions

### 3.1. An Impact of Bacteria on the Growth of Microfungi and White-Rot Fungi

Both the tested fungi and microfungi showed different reactions to the presence of bacteria and *Candida* spp. in their environment (Table 1). Three types of fungi and microfungi behavior were noticed. (i) Some of the tested species even did not start to grow from the beginning of the experiment (value “0” both at the second and ninth days). (ii) Some of the tested species started to inhibit the growth of bacteria by their metabolites at the beginning of the growth, but finally, after several days, bacteria also grew in the previously clear zone, and the fungi or microfungi started to decline. In this case, the clear zone around the tested species varied from 1 to 15 mm on the second day. However, by the ninth day, the clear radius around the mycelium of fungi or microfungi reduced to 0–10 mm. These species tend to grow in height rather than width to avoid contact with bacteria. Eventually, the fungi and microfungi die off. (iii) Some of the tested species grew well, despite the presence of bacteria, because they inhibited bacteria or because both cultures grew independently in symbiosis. Further studies are necessary to determine the exact mechanism of interaction. It was difficult to distinguish the most sensitive specie or strain of the tested fungi and microfungi, as all interactions depended on the strain of fungi/microfungi and bacteria analyzed. Three strains of white-rot fungi were able to grow the mycelium only with the presence of *B. mycoides* in their surroundings, whilst other tested bacteria or *Candida* spp. inhibited the growth of white-rot fungi from the beginning or a few days later. 

For example, *Pleurotus ostreatus* did not grow at all in six cases out of ten, and in three cases *P. ostreatus* slightly inhibited the growth of bacteria on the second day of the experiment, but finally bacteria overgrew these fungi. Microfungi, namely *C. merdarium*, *C. laurentii*, *M. echinate*, *P. paxillin*, *T. harzianum*, and *U. chartarum*, did not grow in the presence of any out of the tested bacteria and yeast (*Candida* spp.), while other microfungi were able to grow with one or six of the tested microorganisms. The growth incompatibility of different species could be explained by the competition due to the nutrients—for example, *A. vinelandii* [58,59] and *K. variicola* [60,61], which are nitrogen-fixing bacteria, or *B. megaterium*, which is a phosphate-solubilizing bacteria [62,63]—or due to different bacteria metabolites, which can inhibit the growth of fungi and microfungi. The literature review shows that some strains of *B. mojavensis* [64], *B. licheniformis* [65,66], *B. subtilis* [67,68], *B. thuringiensis* [12,69,70], and *P. fluorescens* [71,72] exhibit fungicidal characteristics. Even the *B. mycoides* which showed the lowest inhibition rate (six cases out of seventeen) may exhibit nitrogen fixation [73] or antifungal properties [74]. The tested strain of *Candida* spp. isolated form biohumus inhibited the growth of all white-rot fungi and ten out of fourteen species of microfungi (Table 1). As the variety of the genus *Candida* is a largest of medically important yeast [73], their characteristics are also wide, from pathogenic to human [75,76,77] to the use as biocontrol in agriculture by inhibiting other fungi [78,79] or food biotechnology [80]. The detailed mechanism (competition due to nutrients, and inhibition by specific volatile or non-volatile metabolites) of the inhibition of white-rot fungi and microfungi was not under the scope of this study, but it is evident that using a non-sterile substrate for bioremediation may lead to a failure, as white-rot fungi will not grow.

### 3.2. An Impact of Different Chemicals on the Growth of Bacteria, Microfungi, and White-Rot Fungi

By searching for selective chemical agents suppressing substrate microbiota, but not suppressing white-rot fungi as potential bioremediators or usable food stuff, various chemicals were tested. Results of the impact of some inorganic material such as lime, charcoal, dolomite powder, wood ash, gypsum, phosphogypsum, several concentrations of hydrogen peroxide, potassium permanganate, and sodium hydroxide on the growth of microfungi, white-rot fungi, and bacteria are listed in Table 2. Different impact of the used disinfectants was observed. Four levels of inhibition (suppression) were observed during the impact of chemical agents on white-rot fungi, microfungi, and bacteria (Figure 2). Suppression levels varied from weak (±), where growth was inhibited by approximately 10–30%, to full (+++), meaning that 100% growth inhibition was observed during the tested period.

Hydrated lime was selected for the study, as it is used in the agriculture for increasing the pH value and for limewash of trees, and has not only this sun-protective purpose, but also has an antifungal and antiseptic purpose. Souza et al. [8] successfully used hydrated lime solution for the sterilization of substrate for *Pleurotus ostreatus* mushrooms cultivation (substrate was soaked in a 2% solution of hydrated lime). The amount of *E. coli* and total coliforms were reduced in fecal sludge due to increased pH after the addition of lime [81]. However, higher pH can be harmful both to bacteria, microfungi, and white-rot fungi. In our study, both the tested bacteria and microfungi were more strongly suppressed in the impact of lime than white-rot fungi, and unfortunately suppression of white-rot fungi was quite remarkable.

Dolomite was selected for analysis as a natural soil amendment for alkalizing purposes, as it is known that some organisms do not grow at high pH. The measured pH for the additive suspension was around pH 7.5. The dolomite additive did not suppress the growth of any tested microorganism, and even the enhanced growth of microorganisms was observed. The increase in soil microbiota by the amendment of dolomite was determined in the studies of Giagnoni et al. [82] and Malek et al. [23]. A more intensive growth of the tested microorganisms was also observed with charcoal, gypsum, and phosphogypsum. 

Charcoal has shown a positive effect on plant growth, as it increases the soil’s ability to retain plant nutrients and beneficial microbiota, and increases water retention capacity and cation exchange capacity [83]. The use of charcoal as an amendment for diesel oil polluted soil biostimulation allowed us to keep the growth of organotropic bacteria, actinomyces, and fungi [84]. What coincides in our study is that charcoal did not suppressed the growth of bacteria and microfungi. Gypsum amendment is recommended in saline soil, as it reclaims balance of minerals in the soil and increases microbial activity [85]. However, the impact of gypsum on bacteria is dependent on microbiota type and the dose of gypsum [85,86,87]. Phosphogypsum had a positive effect on the growth of tested organisms in our study, which is in agreement with the study of Al-Enazy et al. [88] in saline soil. It is worth mentioning that in the study of Al-Enazy et al. [89], the total count of microorganisms increased by increasing the concentration of phosphogypsum. However, the amendment of calcareous soil with phosphogypsum resulted in the reduction in total microbiota [63]. 

The impact of wood ash to the soil microbiota depends on several factors such as soil origin, ash concentration, incubation time, and type of microorganisms [89]. Ashes can have both positive and negative impacts on the growth of fungi. Wood ash contains nutrients such as potassium, phosphorus, and calcium, which can benefit some species of microorganisms. However, ashes can also be alkaline, and if they raise the pH of the soil too high, this can inhibit the growth of some microorganisms. In our case, weak suppression of microfungi growth was observed after the impact of wooden ashes, while the growth of the tested bacteria was very good (Table 2). Asare-Bediako et al. [90] determined that the suppression of the microfungi by the impact of ashes was both strain- and concentration-dependent. Unfortunately, wood ashes also had a negative impact on the growth of white-rot fungi.

Potassium permanganate and hydrogen peroxide are very popular in medical use as disinfectants and antiseptics. H_2_O_2_ demonstrates broad-spectrum efficacy against viruses, bacteria, yeasts, and bacterial spores [91]. KMnO_4_ is recommended for different skin infections initiated by fungi or bacteria [92], and also can be used as a fungicide to control various types of fungi [93]. It works by releasing oxygen when it comes into contact with fungal spores or mycelium, damaging or killing them. However, the effectiveness of potassium permanganate can vary depending on the concentration, application method, and specific type of fungi involved. Therefore, we decided to evaluate its impact on soil microbiota. Different concentrations both of KMnO_4_ (0.01, 0.1, 0.5, and 1.0%) and H_2_O_2_ (1.5 and 3.0%) were applied to Petri dishes with the tested culture. The tested concentrations of KMnO_4_ did not suppress the growth of microfungi, except *Candida* spp., while H_2_O_2_ possessed strong suppression of microfungi. Tested bacteria were more sensitive to the impact of H_2_O_2_ and KMnO_4_, and even the bactericidal effect was noticed on the second day of treatment. However, the tested white-rot fungi were also inhibited by KMnO_4_, nut not H_2_O_2._

The impact of the tested compounds was dependent on the concentration used and on the specie and strain of bacteria (Table 2). The literature data shows that both potassium permanganate and hydrogen peroxide are mainly used for oxidation of PAH in contaminated soil [94,95] or even in combination with biodegradation [12,96]. A higher concentration (up to 5%) of oxidants was used in the literature, and the total count of bacteria was evaluated. Chen et al. [12] determined if there was a decrease in the total bacteria count from 10^4^ to 10^3^ CFU/g soil by the treatment with 5% hydrogen peroxide for 5 days, then it began to grow slowly after 10 days. It is interesting to note that the changes in microorganism diversity are observed by the impact of different oxidants [96], which means that different species of microorganisms have different revitalization potential or resistance to oxidants. Our data also prove that different bacteria or microfungi show certain behavior to specific oxidants or their concentration.

Sodium hydroxide also had a different impact on the tested microorganisms. It was dependent both on the concentration and on the tested organism. The growth of microfungi was not suppressed by NaOH up to 1.5% concentration (what corresponds pH 13.6). The growth of all microfungi was suppressed after applying 2.0% NaOH (pH 13.7), except *Penicillium funiculosum*, which was growing even after applying 3.0% NaOH (pH 13.9). However, such findings did not prove that the tested organisms were alkaliphilic [97]. This may be explained by the fact that the applied minute amounts of NaOH were partially neutralized when they came into contact with the agar medium in the Petri dishes (pH of agar 6.0–7.5). Probably, in order to achieve a sterilizing effect on NaOH, regular spraying would be necessary. Based on the results obtained, it can be noted that there is no need for a multistep substrate preparation process when H_2_O_2_ is used for selective disinfection of the substrate in order to carry out large-scale white-rot fungi cultivation, i.e., no additional washing or neutralization step is required when applying the H_2_O_2_ solution. This is of utmost importance when dealing with huge amounts of polluted material and industrial-scale remediation, or when producing fungi as a foodstuff. 

## 4. Conclusions

This study shows that white-rot fungi are able to grow with *Bacillus mycoides*, but not other bacteria or yeast (*Candida* spp.) isolated from biohumus, rhizosphere, or hazardous waste such as wooden sleepers. Therefore, it is crucial to achieve initial suppression of the growth of other microbiota, excluding white-rot fungi, in order successfully inoculate and start growing white-rot fungi in natural substrates such as biohumus or its bends, with the waste intended for bioremediation. In the case of bioremediation, revitalized microfungi and bacteria may contribute to the bioremediation process directly or in cooperation with plants used for further bioremediation/composting. The effects of tested chemical agents, i.e., lime (hydrated form), charcoal, dolomite powder, ash, gypsum, phosphogypsum powder, hydrogen peroxide, potassium permanganate, and sodium hydroxide on the growth of microfungi, bacteria isolated from the waste bioremediation substrate, and white-rot fungi were strain dependent. Charcoal, dolomite powder, gypsum, and phosphogypsum could be used for the amendment of the growth media, as both microfungi and bacteria show high viability by the action of these agents. Potassium permanganate has shown to have selective action, and it could be used for bacteria and *Candida* spp. growth suppression, but not for other microfungi. Hydrogen peroxide demonstrated the desired selectivity, and it could be used for the disinfection of the matrix for large-scale cultivation or further waste bioremediation with white-rot fungi, as 1.5 and 3.0% hydrogen peroxide showed strong suppression of microfungi, and the suppression or even the bactericidal effect on some bacteria, but did not suppress the growth of the white-rot fungi. In order to apply the proposed method to large-scale substrate disinfection, the ratio of substrate to H_2_O_2_ should be optimized. Additionally, the depth and homogeneity of the cultivation or bioremediation layer must be considered, as the penetration of H_2_O_2_ into all layers, especially the lower ones, is crucial.

## Figures and Tables

**Figure 1 microorganisms-12-01242-f001:**
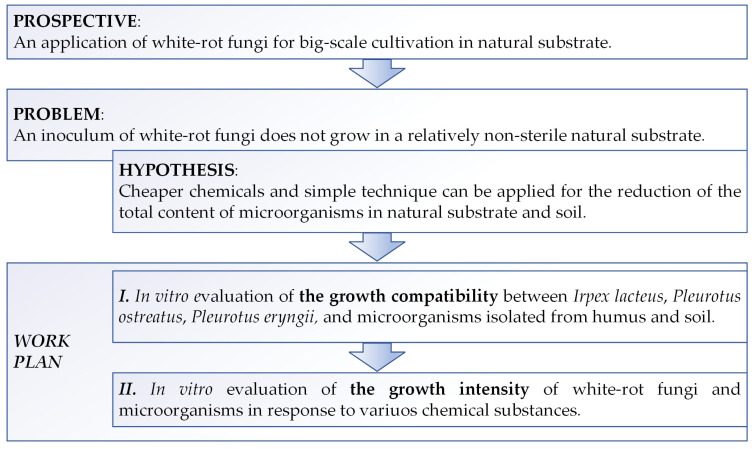
The significance of the work and experimental design.

**Figure 2 microorganisms-12-01242-f002:**
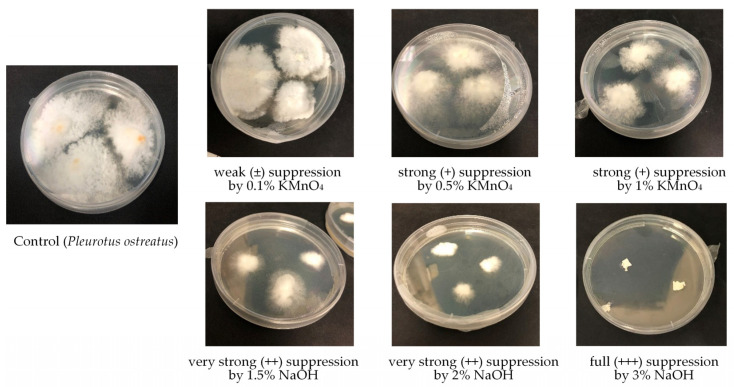
Suppression level of white-rot fungi *Pleurotus ostreatus* by the action of different concentrations of KMnO_4_ and NaOH.

**Table 1 microorganisms-12-01242-t001:** An evaluation of the growth possibilities between bacteria and *Candida* spp. with fungi and microfungi (values represent the radius in mm (±2 mm) of the clear zone around fungi mycelium or inoculum of microfungi, while the radius in mm of newly growing mycelium was measured in the control (without bacteria) sample).

Tested Organism	Control		*A. vinelandii*		*B. licheniformis*		*B. megaterium*		*B. mojavensis*		*B. mycoides*		*B. subtilis*		*B. thuringiensis*		*K. variicola*		*P. fluorescens*		*Candida* spp.	
	Day		Day		Day		Day		Day		Day		Day		Day		Day		Day		Day	
	2nd	9th	2nd	9th	2nd	9th	2nd	9th	2nd	9th	2nd	9th	2nd	9th	2nd	9th	2nd	9th	2nd	9th	2nd	9th
	**White-rot fungi**																					
*I. lacteus*	9	35	8	6 *	7	6	10	8	10	7	3	** 30 ** **	10	8	6	5	6	5	0	0	0	0
*P. ostreatus*	3	25	0	0	0	0	1	0	8	6	0	** 21 **	3	0	0	0	0	0	0	0	0	0
*P. eryngii*	5	30	11	7	7	4	10	8	10	8	2	** 26 **	7	3	5	4	6	5	0	0	0	0
	**Microfungi isolated from the biohumus and used wooden sleepers**																					
*A. verrucosa*	3	17	11	8	5	3	8	7	10	8	4	** 15 **	13	10	6	3	2	** 14 **	0	0	0	0
*A. fumigatus*	5	30	10	8	6	5	8	6	4	2	3	** 24 **	5	0	0	0	0	0	0	** 20 **	0	0
*A. niger*	4	20	5	3	5	0	10	7	0	0	10	** 18 **	4	** 17 **	6	4	1	** 16 **	0	** 15 **	0	** 14 **
*Ch. merdarium*	2	13	8	5	3	0	8	6	5	3	2	0	10	6	7	2	5	4	0	0	0	0
*C. laurentii*	2	8	5	3	2	0	8	5	5	4	0	0	5	4	8	2	5	4	0	0	0	0
*C. neoformans*	2	8	6	4	2	** 4 **	8	6	8	6	0	** 4 **	5	4	5	0	2	0	0	0	0	** 4 **
*F. moniliforme*	6	45	6	4	8	** 38 **	5	3	4	3	4	** 36 **	10	** 34 **	7	** 39 **	2	** 40 **	0	0	0	** 43 **
*M. echinata*	3	20	4	3	3	0	8	6	5	3	2	2	8	4	8	2	3	2	0	0	0	0
*M. verrucaria*	6	30	10	8	0	** 26 **	8	7	5	4	1	** 29 **	8	3	7	5	4	3	0	0	0	** 24 **
*P. funiculosum*	3	28	8	6	2	0	7	5	5	3	4	** 24 **	2	0	4	3	0	0	0	0	0	0
*P. paxilli*	4	23	8	6	0	0	11	9	2	0	0	0	8	3	2	1	0	0	0	0	0	0
*Rh. pusillus*	15	45	10	8	0	** 39 **	8	6	5	3	4	** 40 **	5	** 36 **	2	0	0	** 41 **	0	0	0	0
*T. harzianum*	3	45	4	2	3	2	8	6	5	4	10	9	3	3	6	5	1	0	0	0	0	0
*U. chartarum*	2	15	6	4	4	2	5	3	4	2	0	0	15	8	5	3	4	3	0	0	0	0

* The reduced value on the ninth day means that bacteria inhibit the growth of fungi, and that the growth decline of fungi or microfungi is visible. ** The underlined values show that white-rot fungi or microfungi grow despite the presence of bacteria, and that their mycelium become stronger and expand in width.

**Table 2 microorganisms-12-01242-t002:** The impact of different additives of chemicals * on the growth of white-rot fungi, microfungi, and tested bacteria.

Tested Organism	Lime	Dolomite Powder	Charcoal	Ashes	Gypsum	Phosphogypsum	H_2_O_2_, %		KMnO_4_, %				NaOH, %					
							1.5	3.0	0.01	0.1	0.5	1.0	0.1	0.5	1.0	1.5	2.0	3.0
**White-rot fungi**																		
*Irpex lacteus*	+	++	++	++	+	++	-	-	±	±	+	+	-	-	+++	+++	+++	+++
*Pleurotus ostreatus*	+	++	++	++	+	+	-	-	±	±	+	+	-	-	-	++	++	+++
*Pleurotus eryngii*	+	++	++	++	+	++	-	-	±	±	+	+	-	++	+++	+++	+++	+++
**Microfungi isolated from the biohumus and used wooden sleepers**																		
*Acremoniella verrucosa*	++	-	-	±	-	-	+	+	-	-	-	-	-	-	-	-	++	++
*Aspergillus fumigatus*	++	-	-	±	-	-	+	+	-	-	-	-	-	-	-	-	+	+
*Aspergillus niger*	++	-	-	±	-	-	+	+	-	-	-	-	-	-	-	-	+	+
*Candida* spp.	++	-	-	-	-	-	++	++	++	++	+++	+++	±	±	±	±	±	±
*Chrysosporium merdarium*	++	-	-	±	-	-	+	+	-	-	-	-	-	-	-	-	++	++
*Cryptococcus laurentii*	++	-	-	±	-	-	+	+	-	-	-	-	-	-	-	-	++	++
*Cryptococcus neoformans*	++	-	-	±	-	-	+	+	-	-	-	-	-	-	-	-	++	++
*Fusarium moniliforme*	++	-	-	±	-	-	+	+	-	-	-	-	-	-	-	-	++	++
*Memnoniella echinata*	++	-	-	±	-	-	+	+	-	-	-	-	-	-	-	-	++	++
*Myrothecium verrucaria*	++	-	-	±	-	-	+	+	-	-	-	-	-	-	-	-	++	++
*Penicillium funiculosum*	++	-	-	±	-	-	+	+	-	-	-	-	-	-	-	-	-	-
*Penicillium paxilli*	++	-	-	±	-	-	+	+	-	-	-	-	-	-	-	-	+	+
*Rhizomucor pusillus*	++	-	-	±	-	-	+	+	-	-	-	-	-	-	-	-	++	++
*Trichoderma harzianum*	++	-	-	±	-	-	+	+	-	-	-	-	-	-	-	-	++	++
*Trichophyton rubrum*	++	-	-	±	-	-	+	+	-	-	-	-	-	-	-	-	++	++
*Ulocladium chartarum*	++	-	-	±	-	-	+	+	-	-	-	-	-	-	-	-	++	++
**Bacteria isolated from the biohumus and used wooden sleepers**																		
*Bacillus licheniformis*	++	-	-	-	-	-	++	+++	++	++	++	+++	-	-	-	-	-	-
*Bacillus mycoides*	++	-	-	-	-	-	+++	+++	++	++	+++	+++	-	-	±	±	±	±
*Bacillus subtilis*	++	-	-	-	-	-	+++	+++	++	++	+++	+++	-	-	-	-	-	-
*Bacillus thuringiensis*	++	-	-	-	-	-	+++	+++	++	++	++	++	-	-	-	-	-	-
*Klebsiella variicola*	++	-	-	-	-	-	++	++	++	++	++	+++	-	-	±	+++	+++	+++
*Pseudomonas fluorescens*	++	-	-	-	-	-	++	+++	++	++	+++	+++	-	-	-	-	±	±
**Rhizobacteria from rhizosphere**																		
*Azotobacter vinelandii*	++	-	-	-	-	-	++	++	++	++	++	++	-	±	±	±	±	+++
*Bacillus megaterium*	++	-	-	-	-	-	+++	+++	++	++	++	+++	-	+++	+++	+++	+++	+++
*Bacillus mojavensis*	++	-	-	-	-	-	+++	+++	++	++	++	+++	-	±	+++	+++	+++	+++

* The amount of additives applied on microorganisms spread on a Petri dish: lime, hydrated form, 20 ± 0.1 mg; dolomite powder, 80 ± 0.1 mg; charcoal, 50 ± 0.1 mg; wood ashes, 15 ± 0.1 mg; gypsum, 20 ± 0.1 mg; phosphogypsum powder, 60 ± 0.1 mg; H_2_O_2_ 15 µL; KMnO_4_ 15 µL; NaOH 15 µL. ‘-’ no suppression of the growth of the tested organism by the chemical agent, very good growth of the tested organism; ‘±’ weak suppression (10–30%) of the growth of the tested organism by the chemical agent; ‘+’ strong suppression (30–65%) of the growth of the tested organism by the chemical agent; ‘++’ very strong suppression (65–95%) of the growth of the tested organism by the chemical agent; ‘+++’ full suppression (100%) of the growth of the tested organism by the chemical agent, no growth, bactericidal effect.

## Data Availability

Data supporting reported results are available from the authors.

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
