# Peer review of "Searching for Chemical Agents Suppressing Substrate Microbiota in White-Rot Fungi Large-Scale Cultivation"

_microorganisms, 2024, doi:10.3390/microorganisms12061242_

Round 1

Reviewer 1 Report

Comments and Suggestions for Authors

Edible fungi are a valuable resource in the search for sustainable solutions to environmental pollution. Their ability to degrade organic pollutants, extract heavy metals and restore ecological balance has a huge potential for bioremediation. They are also sustainable food resources. In this study, the author investigated the effects of tested chemical agents i.e., lime (hydrated form), charcoal, dolomite powder, ash, gypsum, phosphogypsum powder, hydrogen peroxide, potassium permanganate and sodium hydroxide on the growth of micromycetes, bacteria isolated from the waste bioremediation substrate and white-rot fungi. And the results showed that hydrogen peroxide demonstrated the desired selectivity and could be suggested for disinfection of the matrix for big scale cultivation or further waste bioremediation with white-rot fungi. It provides a theoretical basis for searching for chemical agents suppressing the growth of micromycetes and bacteria, but not suppressing white-rot fungi. Hence, this work should be published after some major revisions, followed by the comments below:

1. It is suggested that the introduction should be supplemented with research on the inhibition of microbacteria and bacteria by chemical agents.

2. The amount of data in this paper is too small, so it is suggested to add more data.

3. It is recommended to supplement the observation of microstructures of bacteria.

4. In the table, please add specific data on the inhibition of the growth of white-rot fungi, micromycetes and tested bacteria by chemical agents.

5. Please give the picture data of chemical agents inhibiting the growth of white-rot fungi, micromycetes and tested bacteria.

6. Please provide additional evidence that the selected chemical agent inhibits the growth of white-rot fungi, micromycetes and tested bacteria.

Author Response

Dear Reviewer,

Thank you for the review. Here please find the response enclosed.

Yours sincerely, 

Prof. Audrius Maruška

Reviewer 2 Report

Comments and Suggestions for Authors

The submitted manuscript is a fairly well-organized theoretical paper. However, there are several important issues regarding methods and presentation of results, which should be addressed before the work can be published.

It should be noted that all comments, questions and suggestions below are presented with the sole intention of improving the quality of the manuscript for possible publication, and not in a personal capacity.

The different issues to be taken into account and modified are described below:

Line 25: Change "white rot fungi" to "white-rot fungi". This error is made several times throughout the paper. Please correct in all cases. 

Line 56. Add a period after the reference.

Line 162: Avoid shortening the name of the microorganism in this listing/description. 

Line 175: Describe briefly the process to isolate the microorganisms, or refer to any publication related to this process. 

Line 181: Avoid shortening the name of the microorganism in this listing/description. 

Figure 1: The framed descriptions in figure 1 are displaced. Please modify or redo the figure. 

Line 208: Which liquid medium was used? 

Table 2: What is the meaning of "n.t." in table 2? Not tested? If so, why has this test not been performed on the micro-organisms? If possible, incorporate this data or justify why it has not been done. 

Line 378: Include in the conclusion section one or two sentences on future work activities to improve and apply the proposed method. 

Author Response

Dear Reviewer,

Thank you for the review of the manuscript. Here please find the response enclosed. 

Yours sincerely, 

Prof. Audrius Maruška
